# A Vaccine with Multiple Receptor-Binding Domain Subunit Mutations Induces Broad-Spectrum Immune Response against SARS-CoV-2 Variants of Concern

**DOI:** 10.3390/vaccines10101653

**Published:** 2022-10-01

**Authors:** Xu-Chen Hou, Hui-Fang Xu, Yang Liu, Peng Sun, Lin-Wei Ding, Jun-Jie Yue, Tian-Tian Wang, Xin Gong, Jun Wu, Bo Liu

**Affiliations:** 1Department of Microorganism Engineering, Beijing Institute of Biotechnology, Beijing 100071, China; 2School of Mechatronical Engineering, Beijing Institute of Technology, Beijing 100081, China

**Keywords:** SARS-CoV-2, subunit vaccine, VOCs, RBD, yeast

## Abstract

With the emergence of more variants of severe acute respiratory syndrome coronavirus 2 (SARS-CoV-2) and the immune evasion of these variants from existing vaccines, the development of broad-spectrum vaccines is urgently needed. In this study, we designed a novel SARS-CoV-2 receptor-binding domain (RBD) subunit (RBD5m) by integrating five important mutations from SARS-CoV-2 variants of concern (VOCs). The neutralization activities of antibodies induced by the RBD5m candidate vaccine are more balanced and effective for neutralizing different SARS-CoV-2 VOCs in comparison with those induced by the SARS-CoV-2 prototype strain RBD. Our results suggest that the RBD5m vaccine is a good broad-spectrum vaccine candidate able to prevent disease from several different SARS-CoV-2 VOCs.

## 1. Introduction

Since the end of 2019, severe acute respiratory syndrome coronavirus 2 (SARS-CoV-2), the causative agent of coronavirus disease 2019 (COVID-19), has been spreading globally and, as of 10 August 2022, has caused more than 0.58 billion infections and more than 6.4 million deaths (https://covid19.who.int/; accessed on 10 August 2022). SARS-CoV-2 mutates frequently during its transmission, which has resulted in the production of a variety of SARS-CoV-2 variants [1,2]. Because of their demonstrated changes from the prototype strain (PT), such as increased transmissibility or changed clinical disease presentation, the SARS-CoV-2 variants Alpha, Beta, Gamma, Delta, and the newly emerged Omicron are defined as variants of concern (VOCs) by the World Health Organization (WHO). Notably, all these VOCs exhibited different degrees of immune escape from the currently available COVID-19 vaccines [3,4,5,6]. In fact, these vaccines have little effect on preventing infection with the latest variant, Omicron [7]. Thus, it is necessary to develop broad-spectrum vaccines that are more effective in the face of the continued emergence of SARS-CoV-2 variants. However, the rapid evolution of SARS-CoV-2, illustrated by the confirmed ability of the Omicron subvariant BA.4/5 to escape from antibodies induced by the Omicron subvariant BA.1 [8], makes it difficult to keep pace with the antigen mutation using the current vaccine development strategy. Therefore, new strategies for COVID-19 vaccine development should be explored.

The receptor-binding domain (RBD) of spike (S) protein has the specificity to bind with the host receptor angiotensin-converting enzyme 2 (ACE2) of the cell surface, which is the main domain for inducing neutralizing antibodies. For this reason, several RBD-based COVID-19 vaccines and vaccine candidates, such as the Zifivax COVID-19 vaccine and ARCoV, have been developed [9,10]. However, mutations in the RBD were found to be the main cause of immune escape by SARS-CoV-2 variants. For example, the neutralization activity of immune serum induced by current COVID-19 vaccines for pseudoviruses harboring the RBD mutations of Alpha (N501Y), Beta (N501Y/K417N/E484K), Gamma (N501Y/K417T/E484K), and Delta (L452R/T478K) was 1.77, 7.85, 5.12, and 3.0 times lower, respectively, than the neutralization activity of this serum for the SARS-CoV-2 PT [11,12]. Meanwhile, the immune-escaping capability of the single RBD mutation mentioned earlier has also been confirmed [2,6,13,14,15]. Because current vaccine development strategies struggle to keep pace with the speed at which these variants continue to emerge, new strategies for the development of a broad-spectrum COVID-19 vaccine are needed.

To develop a COVID-19 vaccine candidate with the potential for inducing broad-spectrum protection against many SARS-CoV-2 variants, we designed a SARS-CoV-2 RBD subunit, named RBD5m, incorporating five SARS-CoV-2 VOC RBD mutations shown to be important for the immune escape of these variants. We then used our previously developed glycoengineered *Pichia pastoris* expression system and purification process to produce highly purified RBD5m with mammalian N-glycosylation modification. Further experiments clearly demonstrated that RBD5m has strong potential as a broad-spectrum COVID-19 vaccine candidate.

## 2. Materials and Methods

### 2.1. Pichia Pastoris, Bacterial Strains, and Materials

The method used for the construction of the glycoengineered *Pichia pastoris* has been previously reported [16]. The *P. pastoris* strain was cultured in yeast peptone dextrose (YPD) medium at 25 °C. The bacterial strain used *Escherichia coli* Trans5α (TransGen Biotech, Beijing, China), which was cultured in Luria–Bertani (LB) medium at 37 °C. Yeast extract, agar, and tryptone were purchased from Oxoid (Basingstoke, UK); NaCl was purchased from Sinopharm (Shanghai, China); and hygromycin B, zeocin, and G418 were purchased from Thermo Fisher Scientific (Waltham, MA, USA).

### 2.2. Expression, Purification, and Quality Control of the RBD5m Protein

In preliminary work, the SARS-CoV-2 RBD prototype strain (RBDpt) gene (GenBank accession number MN908947.3) was cloned into *Xho*I and *Not*I sites in the pPICZαA vector (Invitrogen, Cal., Carlsbad, CA, USA) to yield pPICZαA-RBDpt [17]. The plasmid pPICZαA-RBDpt was then transferred into the glycoengineered *P. pastoris*, from which RBDpt protein was successfully expressed and purified [17]. The plasmid pPICZαA-RBD5m was designed by integrating multiple RBD mutation sites (K417N, L452R, T478K, E484Q, and N501Y) into the plasmid pPICZαA-RBDpt. After the plasmid pPICZαA-RBD5m was linearized by BglII, it was transformed into the glycoengineered yeast. Expression of the target protein was induced by methanol, and positive clones were identified by screening with sodium dodecyl sulfate–polyacrylamide gel electrophoresis (SDS-PAGE). In accordance with published methods [18], the selected positive strain was fermented in a 5 L bioreactor. The fermentation culture contained 10 g/L of yeast extract (Oxoid), 20 g/L of peptone (Oxoid), 40 g/L of glycerol (Sinopharm), 4.47 g/L of Na_2_HPO_4_ (Sinopharm), 8.22 g/L of NaH_2_PO_4_ (Sinopharm), 3.35 g/L of D-sorbitol (Solarbio, Beijing, China), 5 g/L of (NH_4_)_2_SO_4_ (Sinopharm), 0.4 g/L of MgSO_4_ (Sinopharm), 0.4 g/L of CaCl_2_ (Sinopharm), 1 mL/L of *Pichia* trace mineral (PTM1) salts, and 0.2 mL/L of defoamer agent. For fed-batch fermentation, methyl alcohol (Sinopharm) was supplemented again into the medium after fermentation for 36 h. Methyl alcohol induction was continued for 48–50 h. The fermentation parameters were pH 6.5, temperature 25 °C, and rotation speed 400–800 rpm.

After the fermentation was completed, the product was centrifuged at 8000× *g* rpm for 20 min. The resulting supernatant was purified by being sequentially subjected to a multimodal weak cation exchange (Capto MMC column; GE Healthcare, Cal., Chicago, IL, USA), hydrophobic chromatography (Phenyl Sepharose 6 Fast Flow (low sub); GE Healthcare), a strong anion exchange (Source 30Q column; GE Healthcare), a strong cation exchange (Source 30S column; GE Healthcare), and size-exclusion chromatography (Superdex-75 column; GE Healthcare). At the end of this purification sequence, high-purity RBD5m protein was obtained.

The purity of the RBD5m protein was determined by conducting size-exclusion chromatography–high-performance liquid chromatography (SEC–HPLC) and reverse-phase–high-performance liquid chromatography (RP-HPLC). The absorbance value at 280 nm was recorded (Agilent 1260 HPLC, Cal., Santa Clara, CA, USA). SEC–HPLC (TSK gel G2000SWXL 5 µm Φ 7.8 × 300 mm) was performed using a mobile phase of 20 mM phosphate-buffered saline (PBS) pH 7.0 + 150 mM NaCl at a flow rate of 0.5 mL/min. RP-HPLC was performed using an Agilent ZORBAX 300SB-C8 column (5 µm Φ 4.6 × 250 mm) for analysis. Mobile phase A was 0.1% aqueous trifluoroacetic acid (TFA), and mobile phase B was 0.1% TFA in 95% acetonitrile at a flow rate of 1 mL/min; the elution conditions were 0–80% B for 40 min.

### 2.3. Structural Characterization and Identification of the RBD5m Protein

The purified RBD5m protein was digested by peptide-N-asparagine amidase F (PNGF) at 37 °C. The digested protein was analyzed by performing 12% SDS-PAGE, followed by a Western blot using rabbit serum produced by SARS-CoV-2 RBDpt immunization as the primary antibody and horseradish peroxidase (HRP)-conjugated goat anti–rabbit immunoglobulin G (IgG) antibody (dilution of 1:2000) as the secondary antibody.

### 2.4. Affinity of RBDpt and RBD5m with ACE2

The binding kinetics of RBD5m and RBDpt were determined with the ForteBio Octet™ QKe System (Pall ForteBio Corporation, Cal., Fremont, CA, USA). His-tagged human (h) ACE2 (400 nM) was immobilized on the probe, and then, the RBD5m (400 nM) and RBDpt (400 nM) sample groups and the HBS-EP buffer blank control group were allowed to bind to the hACE2. Next, HBS-EP buffer was used to dissociate the proteins and the dissociation constant (Kd) was determined. The resulting data were analyzed using Data Analysis Software 7.0 (Pall ForteBio Corporation).

### 2.5. Vaccine Preparation and Immunization Detection

A total of 75 female BALB/c mice aged 6–8 weeks (Beijing Weitong Lihua Laboratory Animal Technology Co., Ltd., Beijing, China) were obtained and raised at the Animal Center of the Beijing Institute of Biotechnology. The experimental animal welfare ethics number for our study was IACUC-DWZX-2020-039.

The mice were randomly divided into the following six groups: (1) 10 µg of RBDpt/50 µg of CpG/100 µg of aluminum hydroxide (Al(OH)_3_) gel (Croda, Cowick Hall, UK, 10 µg RBDpt group (*n* = 10); (2) 2.5 µg of RBDpt/50 µg of CpG/100 µg of aluminum hydroxide gel (2.5 µg RBDpt) group (*n* = 10); (3) 10 µg of RBD5m/50 µg of CpG/100 µg of aluminum hydroxide gel (10 µg RBD5m) group (*n* = 10); (4) 2.5 µg of RBD5m/50 µg of CpG/100 µg of aluminum hydroxide gel (2.5 µg RBD5m) group (*n* = 15); (5) 50 µg of CpG/100 µg of aluminum hydroxide gel (CpG + Al(OH)_3_) group (*n* = 15); and (6) the normal saline (NS) group (*n* = 15).

The dual adjuvants CpG and Al(OH)_3_ were added to the mixtures while stirring and then diluted with NS to the required concentrations. The mixed vaccine candidates were stored overnight at 4 °C. The mice in each group were vaccinated twice, once each at days 0 and 14. The mice were vaccinated via a hind-leg intramuscular injection. Blood samples were collected from each mouse on days 0, 14, and 28. At 10 days after the second vaccination, splenocytes were collected from the remaining mice (*n* = 5) in the 2.5 µg RBD5m, CpG + Al(OH)_3_, and NS groups and used to perform an ELISpot trial.

### 2.6. Binding Antibody Titer Measurement

Blood was collected from the immunization animals at 28 days after the first vaccination, and serum was obtained by centrifugation at 8000× *g* rpm for 10 min. Coating buffer (50 mmol/L carbonate, pH = 9.6) was used to dilute the RBDpt, RBDdelta, RBDbeta, RBDomicron, or RBD5m protein to a concentration of 2 µg/mL. Each well of the ELISA plates was filled with 100 µL of the protein-coating buffer mixture and incubated overnight at 4 °C. Phosphate-buffered saline (PBS) containing 0.1% Tween-20 (PBST) was used to wash the plates twice, and then, 300 µL of 5% skim milk (Solarbio) was added to each well and incubated for 1 h at 37 °C. Dilutions (ranging from 1 × 10^5^-fold to 1.024 × 10^7^-fold) of serum separated from the collected blood samples were placed in 11 wells, with two replicates per sample. After the plates had been incubated for 1 h at 37 °C, they were washed three times with PBST. HRP-conjugated goat anti-mouse IgG (dilution of 1:5000) was added to each well. The plates were then incubated for 1 h at 37 °C before being washed four times with PBST. Subsequently, 3,3′,5,5′-tetramethylbenzidine (TMB) single-component substrate solution (Solarbio) was applied for 5–10 min to induce a color reaction. The reaction was terminated by the addition of 50 µL of 2 mol/L H_2_SO_4_, and the results were detected using a 450 nm microplate reader.

### 2.7. Pseudovirus Neutralization Assay

First, serum samples from the experimental mice were diluted in DMEM broth (Gibco, New York, NY, USA) with 10% fetal bovine serum (FBS; ExCell, Shanghai, China) and 1% penicillin–streptomycin solution (Thermo Fisher Scientific), which were then filtered through a 0.22 µm filter. The first well of a 96-well cell plate was filled with 150 µL of the diluted serum. One-third of the serum in the previous well was extracted and added to the subsequent well containing 150 µL of DMEM (i.e., diluted threefold). In total, a dilution series with six different concentrations was prepared. Respectively, 100 µL and 150 µL of DMEM broth were added to the virus control (VC) group and the cell control (CC) group. The pseudoviruses (PT, Beta, Delta, BA.1, BA.2.12.1) were diluted to a concentration of 2 × 10^4^ TCID_50_/m and then added to the sample and VC group wells. The plates were shaken to mix the solutions and then incubated for 1 h at 37 °C. The contents of the wells were transferred to a new 96-well plate that had been filled with 50 µL of HEK293-ACE2 cells (Vazyme, Nanjing, China), containing approximately 2 × 10^4^ cells per well, and the cell plate was then kept in a CO_2_ incubator for 48 h. Finally, the cell plates were stabilized to room temperature and assessed using the Bio-Lite™ Luciferase Assay system (Vazyme). The fluorescent signal was read using a microplate reader.

### 2.8. Tests for Cellular Immunity

Interferon (IFN)-γ, interleukin (IL)-2, IL-4, and IL-5 ELISpot^PLUS^ kits (Mabtech, Stockholm, Sweden) were used to detect IFN-γ, IL-2, IL-4, and IL-5 secretion, respectively, in mouse spleen cells. ELISpot trials were performed by collecting spleen cells at 10 days after the second vaccination from the mice in the 2.5 µg RBD5m immune group, the CpG + Al(OH)_3_ group, and the NS group. The anti-mouse IFN-γ, IL-2, IL-4, or IL-5 antibody pre-coated strip plates were washed four times with sterile PBS. Next, 200 µL of blocking buffer (RPMI 1640 solution + 10% FBS + 1% penicillin–streptomycin solution) was added to each well for incubation at room temperature for at least 30 min. After removing the medium from the ELISpot plates, 100 µL of the following were added to the plate well: dimethyl sulfoxide (DMSO) for the negative control group; SARS-CoV-2 S protein full-length peptide pool (2 µg/mL) for the experimental groups; 8 µg/mL of ConA for the positive control group on the IFN-γ, IL-2, or IL-4 plate; and 100 ng/mL PMA + 1 µg/mL of ionomycin for the positive control group on the IL-5 plate. Next, 2 × 10^5^ cells/100 µL (IL-5 plates: 5 × 10^5^ cells/100 µL) of splenocytes were added to each well, and the plates were incubated at 37 °C in a 5% CO_2_ incubator for 36 h. The solutions in the plates were then removed, and the plates were washed five times with PBS. Biotinylated anti-mouse IFN-γ, IL-2, IL-4, or IL-5 monoclonal detection antibodies were diluted to 1 µg/mL in PBS with 0.5% FBS (PBS-0.5% FBS). Next, 100 µL of the appropriate antibody solution was added to each well of the plates. After the plates had been incubated for 2 h at room temperature, they were washed five times with PBS. Streptavidin–ALP was diluted 1:1000 times in PBS-0.5% FBS, added to the wells, and incubated with the plates at room temperature for 1 h. After the plates had been washed five times with PBS, 100 µL of the substrate solution BCIP/NBT-plus (filtered by a 0.45 µm filter) was added to each well and left to react for 2–15 min until distinct spots emerged. Afterward, the plates were washed extensively with deionized water. After the plates had fully dried, the spots were scanned and counted using an ELISpot reader.

## 3. Results

### 3.1. Design of the COVID-19 Vaccine Candidate by Integrating Five RBD Mutations

To develop a broad-spectrum COVID-19 vaccine, we designed a mutated RBD, named RBD5m, by introducing five mutations that are able to cause immune escape: K417N, L452R, T478K, E484Q, and N501Y (Figure 1A). K417N and N501Y were derived from the mutation sites of the Beta variant RBD; L452R and E484Q were derived from the mutation sites of the Kappa variant RBD; and T478K was derived from the mutation sites of the Delta variant RBD. The coexistence of L452R and E484Q enhanced infectivity, while blemishing the binding reaction between S protein and the antibody generated upon vaccination, thus enhancing the immune escape [19]. The configuration of RBD5m was simulated with PyMol software, as shown in Figure 1B. The newly designed RBD was expressed by a glycoengineered *P. pastoris* expression system we had developed previously [18]. In comparison with previous *P. pastoris* expression systems, the novel one we adopted in this study enables the foreign protein to undergo mammalian N-glycosylation modification, which usually results in better immunogenicity and has no influence on the high expression level of the protein.

### 3.2. Purified RBD5m Has Biological Activities

The expression of RBD5m by glycoengineered *Pichia pastoris* was first verified by SDS-PAGE. As shown in Figure 2A, a protein band between 25 and 35 kDa in size was found, which is in accordance with the expected protein molecular weight of the glycosylated RBD (30 kDa). After treatment with peptide-N-glycosidase F (PNGF), RBD5m shifted to a protein with a molecular weight in the range of 17–25 kDa, which corresponds to the calculated molecular weight of de-glycosylated RBD5m (Figure 2A). Western blot results further verified the successful expression and glycosylation modification of RBD5m (Figure 2A). After that, the expressed RBD5m was purified in accordance with a standard procedure we developed previously [18]. The purity of the purified RBD5m was analyzed using SEC-HPLC and RP-HPLC. As shown in Figure 2B and Appendix A, the purity of the obtained RBD5m was higher than 98%. The binding kinetics between RBDpt or RBD5m and hACE2 were determined by ForteBio. The expressed RBD5m retained an affinity for hACE2, similar to that of RBDpt, as demonstrated by kinetics analysis data (Figure 2C, Appendix A). These results indicate that the biological activities of RBD5m are similar to those of RBDpt.

### 3.3. RBD5m Induces RBD-Specific Binding Antibodies in Mice

We evaluated the immunogenicity of the purified antigen RBD5m in BALB/c mice in comparison with that of RBDpt. CpG (CpG2006) and aluminum hydroxide gel were used as dual adjuvants for RBD5m or RBDpt vaccination. Initially, groups of mice were intramuscularly vaccinated with RBD5m or RBDpt at a dose of 2.5 µg (low-dose groups) or 10 µg (high-dose groups), and another group vaccinated with the dual adjuvants (CpG + Al(OH)_3_) or normal saline (NS group) was set as a placebo control group. On day 14 post-vaccination, the immunized groups were boosted with a dose identical to that used for the initial vaccination (Figure 3A). During the observation period, no local inflammation response at the injection site, bodyweight loss (Appendix A), or other visible symptoms were found. As our previous studies have shown that the titers of RBDpt-induced antibodies peak on approximately day 28 after the primary vaccination [18], we collected serum samples from the vaccinated mice at the same time point for determining the titers of SARS-CoV-2 RBD-specific binding (IgG) antibodies and neutralization antibodies (nAb). Furthermore, IFN-γ, IL-2, IL-4, and IL-5 expression was detected in mouse splenocytes on day 24 after the primary vaccination.

The titers of SARS-CoV-2 RBD-specific binding antibodies induced by RBDpt and RBD5m against the RBDs of SARS-CoV-2 PT, SARS-CoV-2 VOCs, or RBD5m protein were determined by ELISA. As shown in Figure 3B and 3C, both RBDpt and RBD5m induced the production of RBD-specific binding antibodies at both immunization doses. At the low immunization dose, the antibodies against the RBD of SARS-CoV-2 PT, SARS-CoV-2 Beta, SARS-CoV-2 Delta, or RBD5m induced by RBDpt exhibited binding affinities that were similar to those induced by RBD5m (Figure 3B). At the high immunization dose, there was also no difference between the binding affinities of antibodies induced by RBDpt and that of antibodies induced by RBD5m for the RBDs of different virus strains. Furthermore, the induced antibodies similarly displayed comparable levels of binding affinity for each of the four tested RBDs (Figure 3C). The results revealed that there were no differences between RBDpt and RBD5m in binding antibody titers.

### 3.4. RBD5m Elicits a Broad-Spectrum Immune Response in Mice

We tested the neutralization titers of the antibodies induced by RBD5m or RBDpt for SARS-CoV-2 PT, Beta, Delta, and Omicron subvariants BA.1 and BA.2 (BA.2.12.1) with corresponding SARS-CoV-2 pseudoviruses. As shown in Figure 4A, the neutralization titers of low-dose RBDpt-induced antibodies for SARS-CoV-2 PT, Beta, Delta, BA.1, and BA.2 approached approximately 523, 284, 441, 37, and 30 median effective dose (ED_50_), respectively. In comparison, the neutralization titers of low-dose RBD5m-induced antibodies for SARS-CoV-2 PT, Beta, Delta, BA.1, and BA.2 approached approximately 272, 368, 528, 388, and 391 ED_50_, respectively. Statistical analysis revealed that the variation among the neutralization titers of the RBDpt-induced antibodies for the PT, Beta, Delta, BA.1, and BA.2 pseudoviruses was significant, whereas the variation among the neutralization titers of the RBD5m-induced antibodies for the PT, Beta, Delta, BA.1, and BA.2 pseudoviruses was not significant. Notably, the RBDpt-induced antibodies almost failed to neutralize BA.1 and BA.2, while the RBD5m-induced antibodies reached high neutralization levels for these two Omicron subvariants and showed comparable neutralization for the Beta and Delta variants. Similar results were also found with high-dose vaccination (Figure 4B). The neutralization titers of high-dose RBDpt-induced antibodies for SARS-CoV-2 PT, Beta, Delta, BA.1, and BA.2 approached approximately 868, 364, 591, 55, and 32 ED_50_, respectively; high-dose RBDm5-induced antibodies approached approximately 250, 226, 1195, 2543, and 1102 ED_50_, respectively.

From these results, we surmise that RBD5m vaccination induces broad-spectrum antibodies in contrast to RBDpt vaccination, indicating that RBD5m is an ideal universal vaccine candidate for preventing disease from multiple SARS-CoV-2 variants.

### 3.5. RBD5m Effectively Elicits a T Cell Immune Response

The T cell response is also an important indicator for assessing SARS-CoV-2 vaccine development [20]. To quantify the specific T cell subsets induced by antigens, ELISpot trials detecting interferon (IFN)-γ and interleukin (IL)-2 secreted by T-helper-1 (Th1) cells as well as IL-4 and IL-5 secreted by T-helper-2 (Th2) cells were performed on the splenocytes of vaccinated mice. As shown in Figure 5, vaccination with even low-dose RBD5m induced higher levels of IFN-γ-, IL-2-, IL-4-, and IL-5-expressing cells in response to ex vivo stimulation with the full-length SARS-CoV-2 S protein peptide pool compared to the levels of these cells in the adjuvant and NS groups. These results indicate that the RBD5m vaccine effectively activated obvious antigen-specific Th1 and Th2 cell responses.

## 4. Discussion

The RBD of S protein as the antigen of the SARS-CoV-2 vaccine can induce the immunological system to produce specific antibodies against the RBD region, thus preventing virus infection. In this study, we designed a novel SARS-CoV-2 RBD subunit (RBD5m) by introducing five hallmark substitutions from different SARS-CoV-2 VOCs or variants of interest that have been confirmed to be capable of causing immune evasion. Vaccination with the purified RBD5m induced a high level of neutralizing antibodies, which not only exhibited robust neutralization for the previously dominant SARS-CoV-2 strains, i.e., SARS-CoV-2 PT, Beta, and Delta, but also showed similarly robust neutralizing activity for the recently emerged Omicron subvariants BA.1 and BA.2. Meanwhile, the RBD5m vaccine elicited secreting IFN-γ and IL-2 by Th1 immune cells and IL-4 and IL-5 by Th2 immune cells. Therefore, RBD5m vaccine has the ability to effectively induce humoral and cellular immune responses.

The currently available COVID-19 vaccines were all designed or created from the PT SARS-CoV-2, and they are now failing to prevent disease from the circulating SARS-CoV-2 variants. More importantly, the currently circulating SARS-CoV-2 variant Omicron is still rapidly mutating. Omicron subvariants that appeared later have been demonstrated as being capable of escaping from the antibodies induced by the Omicron subvariants that appeared earlier; consequently, the current strategy of SARS-CoV-2 vaccine development struggles to keep pace with viral mutation. Our specially designed subunit vaccine candidate, RBD5m, provides a better chance of keeping up with the rapid evolution of SARS-CoV-2. In our study, the newly designed RBD was successfully expressed by a special *P. pastoris* expression system we built previously. This expression system not only guaranteed the high expression of RBD5m but also provided a protein product modified with mammalian glycosylation, which is important for maintaining the immunogenicity of the antigen. Additionally, the purity of our RBDm5 protein was over 98% after completing a purification process that followed a standard procedure we built in our laboratory. The results indicate that we not only generated a potential broad-spectrum subunit vaccine candidate for COVID-19 but also constructed a mature technical process for antigen production, which further demonstrates the potential for the clinical application of RBD5m.

## 5. Conclusions

As current vaccine development strategies struggle to keep up with the rate of SARS-CoV-2 variants’ mutation, new strategies for the development of a broad-spectrum COVID-19 vaccine are urgently needed. In this study, we designed a COVID-19 vaccine with multiple RBD subunit mutations, named RBD5m. According to our results, a high-purity RBD5m protein (purity > 98%) was reported to have the same affinity with the hACE2 as the RBD of the SARS-CoV-2 PT. BALB/c mice vaccinated twice with the RBD5m candidate vaccine containing dual adjuvants (CpG and Al(OH)_3_) developed high binding antibody titers. Compared to the antibodies induced by vaccination of mice with the RBD of SARS-CoV-2 PT, the antibodies induced by RBD5m vaccination were more effective in neutralizing a broad range of SARS-CoV-2 VOCs (SARS-CoV-2 PT, Beta, Delta, Omicron BA.1, and Omicron BA.2). Importantly, the RBD5m candidate vaccine produced by the glycoengineered *Pichia pastoris* platform has the advantages of high expression and provides the protein product modified with mammalian glycosylation, which is conducive to factory mass production. Therefore, not only can the RBD5m vaccine development strategy used in this research provide a useful approach for further vaccine development but also the candidate vaccine tested in this research has exciting potential for clinical utility.

## Figures and Tables

**Figure 1 vaccines-10-01653-f001:**
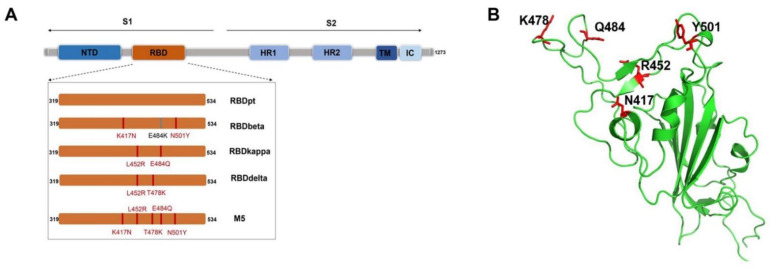
Design of the RBD5m protein. (**A**) Schematic illustration of the RBD5m design. The mutant sites of the receptor-binding domains (RBDs) of the SARS-CoV-2 Beta (RBDbeta), Kappa (RBDkappa), and Delta (RBDdelta) variants are marked in red or gray. In the RBDs of the SARS-CoV-2 variants, the amino acid sites overlapping with the RBD5m protein mutation sites are marked in red and the non-overlapping sites are displayed in gray. (**B**) Structure of RBD5m, as predicted by the PyMol simulation. The five amino acid sites of RBD5m that differ from RBDpt are highlighted in red.

**Figure 2 vaccines-10-01653-f002:**
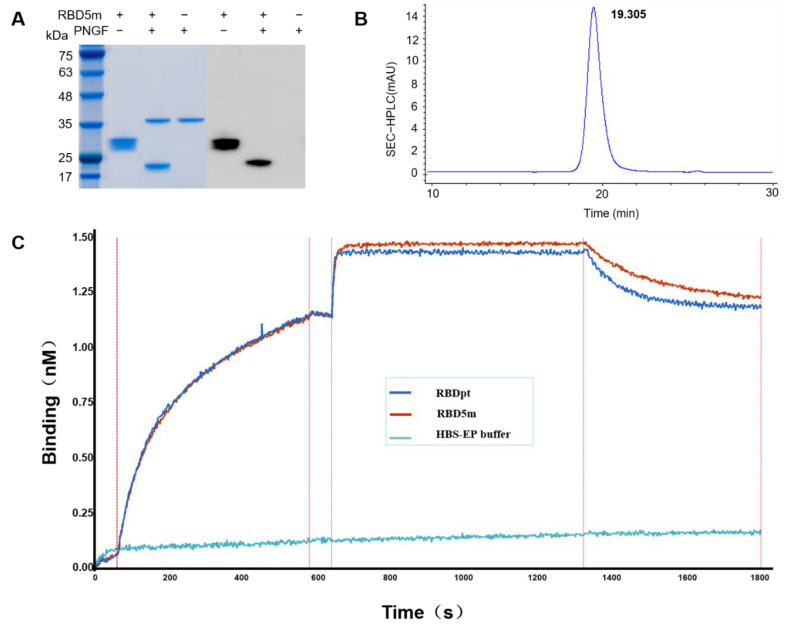
Characterization of RBD5m. (**A**) SDS-PAGE and Western blot analyses of the undigested or PNGF-digested RBD5m protein. (**B**) Purity analysis of the purified RBD5m, conducted with SEC–HPLC. (**C**) Analysis of the RBDpt and RBD5m affinity to ACE2.

**Figure 3 vaccines-10-01653-f003:**
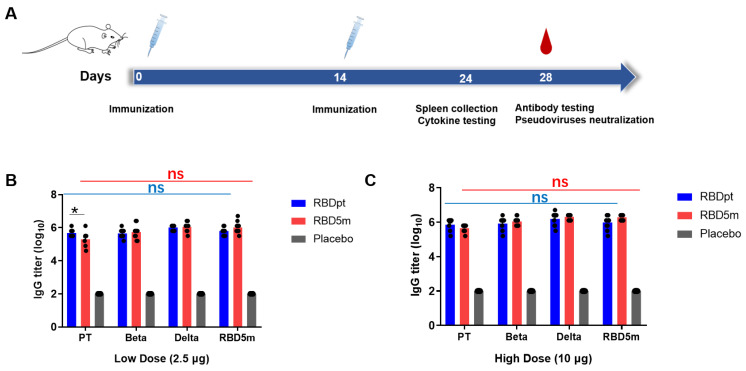
SARS-CoV-2 RBD-specific binding antibody titers. (**A**) Schematic diagram of immunization and serum sample collection. BALB/c mice were immunized at days 0 and 14 via an intramuscular injection at a volume of 100 μL. Blood samples were collected at day 28 after the primary vaccination for use in an RBD-specific binding antibody titer test and a neutralization activity analysis. (**B**,**C**) RBD-specific IgG antibody titers of the low-dose (2.5 µg) groups (**B**) or high-dose (10 µg) groups (**C**) of RBDpt and RBD5m against the RBDs of the prototype strain (PT), Beta, and Delta SARS-CoV-2 and against RBD5m, as determined by using ELISA. ns: no significant difference; * *p* < 0.05.

**Figure 4 vaccines-10-01653-f004:**
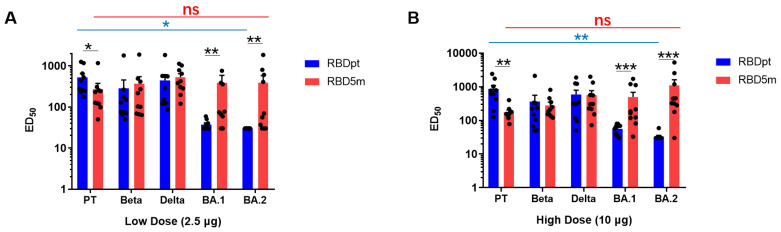
RBD5m elicits broad-spectrum neutralizing antibodies in mice. (**A**,**B**) Specific neutralization titers of the low-dose (2.5-µg) groups (**A**) or high-dose (10-µg) groups (**B**) of RBDpt and RBD5m against SARS-CoV-2 prototype strain (PT), Beta, Delta, BA.1, and BA.2 were determined by using pseudoviruses. ns: no significant difference; * *p* < 0.05; ** *p* < 0.01; *** *p* < 0.001; ED_50_: median effective dose.

**Figure 5 vaccines-10-01653-f005:**
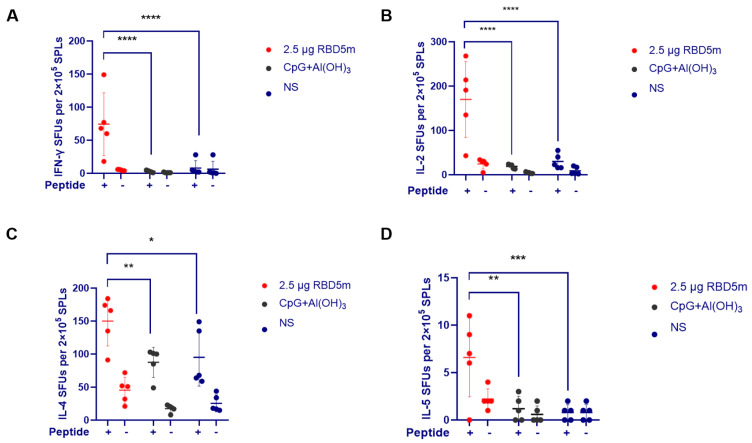
RBD5m immunization elicits a T cell response in BALB/c mice. (**A**–**D**) IFN-γ (**A**), IL-2 (**B**), IL-4 (**C**), and IL-5 (**D**) expression was detected in mouse splenocytes by ELISpot trials. The “+” sign indicates that mouse splenocytes were stimulated by the addition of the full-length peptide for SARS-CoV-2 S protein. The data were collected on an ELISpot board reader. ns: no significant difference; * *p* < 0.05; ** *p* < 0.01; *** *p* < 0.001; **** *p* < 0.0001.

## Data Availability

Not applicable.

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
