# Peer review of "A Vaccine with Multiple Receptor-Binding Domain Subunit Mutations Induces Broad-Spectrum Immune Response against SARS-CoV-2 Variants of Concern"

_vaccines, 2022, doi:10.3390/vaccines10101653_

Round 1
Reviewer 1 Report
In this manuscript, Hou, Xu, et al. designed a novel SARS-CoV-2 RBD subunit immunogen, RBD5m, incorporating five mutations from variants of concern, with the purpose of inducing broad-spectrum immune responses. The protein immunogen was generated in an engineered yeast expression system with proper glycan modifications, and was utilized for mouse immunization in comparison with the corresponding wild-type version of RBD. Though no difference in binding profiling, RBD5m showed superior neutralizing antibody titers especially against Omicron variants. Cellular immunity was also properly elicited by this immunogen. The incorporation of the VOC mutations into RBD immunogen design can provide a potential direction for future vaccine development.
There are several points in this manuscript that need to be addressed:
· In the title the authors indicate the vaccine provide broad-spectrum “protection” against VOCs, however, there is no in vivo virus challenge data in this manuscript that support the word “protection”. So unless the authors include the animal protection data, this word might not be proper to use in the title, or the authors may use “induce broad-spectrum immune response” instead.
· In line 109, the sub-title of section 2.3 was “structural characterization of RBD5m”, but this description may not be proper for western blot assays.
· In section 3.1, the authors may explain more in detail why these five mutations in RBD were chosen, as well as cite some related references indicating the immune escape they have caused.
· Was there any glycan analysis of the produced protein to support that the glycoengineered yeast expression platform adopted in this study is functioning as expected? Especially site-specific glycan analysis, since the glycan forms generated in mammalian cells and yeast are different.
· In Table S1 which showed Ka, Kd, and KD values, all the values for both RBDwt and RBD5m are exactly the same, could the authors double check whether this is true?
· In Figure 3, where are the dots for placebo group? In addition, the color of dots of RBDwt and RBD5m groups can be the same (instead of black and grey), since the bar colors are already different, therefore the dot colors are redundant information (for both Figure 3 and Figure 4).
· In section 3.4, did the authors test the neutralization against other VOCs including alpha, gamma, etc. as well as other new BA variants (BA.2.12.1, BA.4/5, BA.2.75)?
· Based on the neutralizing antibody results, there is lower titers against WT for the RBD5m immunized group, could the authors add 1-2 more sentences commenting on that?
· For the cellular immunity analysis, why the authors didn't include the RBDwt group as well for better comparison? In addition, since RBD subunit was used as the immunogen, why did the authors use the spike protein full-length for the peptide pool for T helper cell stimulation? And in Figure 5, why did the negative group show some levels of IL-4 expression?
· In line 369, there is a typo “provid”.
Author Response
Response to Reviewer 1 Comments
Point 1: In the title the authors indicate the vaccine provide broad-spectrum “protection” against VOCs, however, there is no in vivo virus challenge data in this manuscript that support the word “protection”. So unless the authors include the animal protection data, this word might not be proper to use in the title, or the authors may use “induce broad-spectrum immune response” instead.
Response 1: We gratefully appreciate for your valuable comment. And we have been revised in the manuscript.
Point 2: In line 109, the sub-title of section 2.3 was “structural characterization of RBD5m”, but this description may not be proper for western blot assays.
Response 2: Thank you for your rigorous consideration. We have changed the sub-title to “Structural characterization and identification of the RBD5m protein”.
Point 3: In section 3.1, the authors may explain more in detail why these five mutations in RBD were chosen, as well as cite some related references indicating the immune escape they have caused.
Response 3: Thank you for pointing this out. We have revised sexction 3.1 in our manuscript.
Point 4: Was there any glycan analysis of the produced protein to support that the glycoengineered yeast expression platform adopted in this study is functioning as expected? Especially site-specific glycan analysis, since the glycan forms generated in mammalian cells and yeast are different.
Response 4: In our previous work (reference 18), we analyzed the N-glycans of the RBDwt. The result showed that RBDwt expressed by glycoengineered platform mainly possess complex-type N-glycans(GlcNAc2Man3GlcNAc2) and have a small amount of mannose N-glycans (Man5GlcNAc2) without phosphorylation, which differed from the high-mannose-type glycan chains expressed by wild-type yeast cells.The RBD5m protein treated with PNGF was consistent with theoretical mass of 25 kDa, suggesting that the RBD5m had undergone N-glycosylation modification. Therefore, we preliminarily assumed that RBD5m has similar glycosylation modifications as RBDwt. They have mammalian-like glycosylation that makes the vaccine more suitable for humans.
Point 5: In Table S1 which showed Ka, Kd, and KD values, all the values for both RBDwt and RBD5m are exactly the same, could the authors double check whether this is true?
Response 5: Thank you so much for your careful check. We have confirmed that the data in Table S1 is ture.
Point 6: In Figure 3, where are the dots for placebo group? In addition, the color of dots of RBDwt and RBD5m groups can be the same (instead of black and grey), since the bar colors are already different, therefore the dot colors are redundant information (for both Figure 3 and Figure 4).
Response 6: Thank you for your nice suggestion. And we have modified Figure 3 and 4.
Point 7: In section 3.4, did the authors test the neutralization against other VOCs including alpha, gamma, etc. as well as other new BA variants (BA.2.12.1, BA.4/5, BA.2.75)?
Response 7: It is regret that we did not perform so broad netrulization experiments. According to previous studies, the variant Beta, Delta and Omicron variants showed more serious immune escape, so we only foucused on these three variants. BA.2 variant mentioned in the manuscript is BA.2.12.1, which we have revised in section 2 and section 3. However, the other new BA variants such as BA.4/5 and BA.2.75 did not become the pandemic strains when we submit the manuscript, so we didn’t test the neutralization of these variants.
Point 8: Based on the neutralizing antibody results, there is lower titers against WT for the RBD5m immunized group, could the authors add 1-2 more sentences commenting on that?
Response 8: The neutralizition titers induced by RBD5m for SARS-CoV-2 WT pseudovirus is lower than that of RBDwt, may be because RBD5m has five different mutation sites from RBD of WT pseudovirus, while the mutation sites of RBDwt completely coincident with RBD of WT pseudovirus. The five mutation sites of RBD5m resulted in decreased affinity for binding to SARS-CoV-2 WT pseudovirus.
Point 9: For the cellular immunity analysis, why the authors didn't include the RBDwt group as well for better comparison? In addition, since RBD subunit was used as the immunogen, why did the authors use the spike protein full-length for the peptide pool for T helper cell stimulation? And in Figure 5, why did the negative group show some levels of IL-4 expression?
Response 9: Thanks a lot for the reviewer’s comments. And we have address each point:
- Our study is mainly foucused on the RBD5m candidate vaccine, so RBDwt groups is not the point of the experiment.
- The reason of using the full-length S protein peptide pool for T-cell stimulation is to imitate the natually occurring SARS-CoV-2 viruses.
- It seems that it is a common reaction for the negative group to show some levels of IL-4 expression after using the peptide pool to stimulate the T helper cells, as shown in other similar studies1.
- Na-Na Zhang et al. A Thermostable mRNA Vaccine against COVID-19. Cell 182, 1271-1283
Point 10: In line 369, there is a typo “provid”.
Response 10: We are very sorry for our careless mistake and it was rectified at Line 369.
Reviewer 2 Report
This article is very interesting and in general, is well-written and designed. However, there are some minor aspects that need to be addressed:
1. Line 28. The term WT virus is not the correct one. All naturally occurring variants are wild-type. Perhaps the best way to describe it would be “the original variant” or “the original Wuhan sequence”.
2. The introduction is fine.
3. Line 72. What do you mean by “in preliminary work”?
4. Section 2.2 is written in a very unclear way. Especially the first 8 lines.
5. Section. nM/L is not a correct concentration unit. Do you mean nanomoles / L if so then write nM.
6. Line 144. The proper term is immunization not vaccination.
7. Section 2.7. What kind of pseudovirus was used? Same comment for section 3.4
8. Lines 213-216. What do you mean by the novel? Please explain why your system is novel and how was this achieved. I guess you mean that this P. Patty can do N-glycosylation, which is described in reference 18, but it will help to briefly explain this system.
Author Response
Response to Reviewer 2 Comments
Point 1: Line 28. The term WT virus is not the correct one. All naturally occurring variants are wild-type. Perhaps the best way to describe it would be “the original variant” or “the original Wuhan sequence”.
Response 1: We appreciate it very much for this good suggestion, and we have done it according to your ideas. It has been revised in the manuscript.
Point 2: The introduction is fine.
Response 2: Thank you very much for the nice comments!
Point 3: Line 72. What do you mean by “in preliminary work”?
Response 3: The preliminary work have been reported globally (reference 18). In preliminary work, the plasmid of pPICZαA-RBDwt have been constructed and the protein of RBDwt have been expressed and purified.
Point 4: Section 2.2 is written in a very unclear way. Especially the first 8 lines.
Response 4: We feel sorry for the inconvenience brought to the reviewer. We have revised in Section 2.2 of the manuscript.
Point 5: Section. nM/L is not a correct concentration unit. Do you mean nanomoles / L if so then write nM.
Response 5: Thanks for reviewer’s comments and it was rectified in section 2.4.
Point 6: Line 144. The proper term is immunization not vaccination.
Response 6: We are very sorry for our careless mistake and it was rectified at line 144.
Point 7: Section 2.7. What kind of pseudovirus was used? Same comment for section 3.4
Response 7: Thank you so much for your careful check. And we have rectified in section 2.7 and 3.4.
Point 8: Lines 213-216. What do you mean by the novel? Please explain why your system is novel and how was this achieved. I guess you mean that this P. Patty can do N-glycosylation, which is described in reference 18, but it will help to briefly explain this system.
Response 8: Thanks for reviewer’s comments. N-glycosylation in previous P. pastoris expression systems leads to hyper-mannosylation, which will affecting the immunogenicity and protective effect of the vaccines. Neverthless, the novel P. pastoris expression system with characteristics of glycosylation modification similar to those of mammalian cells which have good immunogenicity and protective effects. To achieve this purpose, we have knocked out or inserted the N-glycosylation-related genes in glycoengineered Pichia pastoris in the preliminary work at our laboratory.